# Whole-Exome Sequencing in Family Trios Reveals De Novo Mutations Associated with Type 1 Diabetes Mellitus

**DOI:** 10.3390/biology12030413

**Published:** 2023-03-07

**Authors:** Mira Mousa, Sara Albarguthi, Mohammed Albreiki, Zenab Farooq, Sameeha Sajid, Sarah El Hajj Chehadeh, Gihan Daw ElBait, Guan Tay, Asma Al Deeb, Habiba Alsafar

**Affiliations:** 1Center of Biotechnology, Khalifa University of Science and Technology, Abu Dhabi 127788, United Arab Emirates; 2Department of Biomedical Engineering, Khalifa University of Science and Technology, Abu Dhabi 127788, United Arab Emirates; 3College of Medicine and Health Sciences, Khalifa University, Abu Dhabi 127788, United Arab Emirates; 4Department of Endocrinology, Mafraq Hospital, Abu Dhabi 127788, United Arab Emirates

**Keywords:** type 1 diabetes, WES, case–parent trios, de novo variants, family trios, T1D, diabetes

## Abstract

**Simple Summary:**

Type 1 diabetes mellitus (T1DM) is a chronic autoimmune condition in which the immune system destroys insulin-making cells in the pancreas. Many advances have been made in the past decade to understand the pathophysiology of T1DM. With an estimated heritability risk of 50%, the strong genetic component plays an important role in the discovery of novel disease pathways and identification of new targets for therapeutic purposes. In this study, we aim to identify new (de novo) genetic markers for T1DM patients by sequencing the genes of the affected individual and their parents (trio family). This is a powerful approach to identify causal mutations for inherited diseases, such as T1DM, to improve our understanding of the condition. With 13 trio families, we identified 32 new (de novo) genetic mutations. Of these, 12 variants that were linked to T1DM, and the remaining 20 variants were linked to endocrine, metabolic, or autoimmune diseases. The findings of this study have allowed us to identify the genetic markers associated with the development of T1DM, to be able to improve diagnosis through therapeutic advancements.

**Abstract:**

Type 1 diabetes mellitus (T1DM) is a chronic autoimmune disease characterized by insulin deficiency and loss of pancreatic islet β-cells. The objective of this study is to identify de novo mutations in 13 trios from singleton families that contribute to the genetic basis of T1DM through the application of whole-exome sequencing (WES). Of the 13 families sampled for this project, 12 had de novo variants, with Family 7 having the highest number (nine) of variants linked to T1DM/autoimmune pathways, whilst Family 4 did not have any variants past the filtering steps. There were 10 variants of 7 genes reportedly associated with T1DM (*MST1; TDG; TYRO3; IFIHI; GLIS3; VEGFA; TYK2*). There were 20 variants of 13 genes that were linked to endocrine, metabolic, or autoimmune diseases. Our findings demonstrate that trio-based WES is a powerful approach for identifying new candidate genes for the pathogenesis of T1D. Genotyping and functional annotation of the discovered de novo variants in a large cohort is recommended to ascertain their association with disease pathogenesis.

## 1. Introduction

Type 1 diabetes mellitus (T1DM) is a chronic autoimmune disease primarily characterized by insulin deficiency and loss of pancreatic islet β-cells [1,2]. T1DM accounts for 5–10% of total cases of diabetes worldwide, and is one of the most common endocrine and metabolic conditions occurring in childhood. The peak age of presentation of childhood-onset T1DM has a bimodal distribution, with the first peak between four to six years, and the second in early puberty [3]. The pathogenesis of T1DM has been suggested to be a continuum development, starting from the detection of autoantibodies before symptom onset to the progression of β-cell destruction, dysglycaemia, and hyperglycemia [4]. 

T1DM is a heterogeneous multifactorial condition, and the elucidation of its complex etiology is highly dependent on the interaction between numerous environmental factors precipitated by genetic susceptibility. Environmental influences, such as lifestyle, viral infections (especially respiratory such as rotavirus, cytomegalovirus, and mumps or coxsackie B and other enterovirus infections), and gestational events have been proposed as candidate etiological factors. In individuals with genetic predisposition, exposure to these environmental triggers leads to an autoimmune response, mediated by autoreactive CD4 and CD8 T-cells, resulting in the destruction of insulin-producing beta cells [5]. 

Family studies have revealed that T1DM is heritable, with an estimated 50% risk. The concordance rate among monozygotic twins is reported to be 30% within 10 years of diagnosis of the first twin, reaching as high as 65% by 60 years of age [6]. A child of an affected mother has a lifetime rise between 1 and 4%, whereas a child of an affected father has a 3–8% risk, and where both parents are affected, the risk reaches up to 30% [7]. Linkage and genetic association studies have identified >50 loci that contribute to susceptibility or resistance to developing T1DM, including its association with pancreatic beta cell autoimmunity (insulin gene (INS)), inflammatory-associated factors (interleukin-2 signaling pathway, B-cell and T-cell development, and cytokine signaling), human leucocyte antigen (HLA) genes, and shared genetic architecture with autoimmune diseases (APS1 and STAT3 poly-autoimmunity) [8,9,10].

Despite the rapid advancements in genetic methods, our ability to understand the pathogenesis of T1DM to improve therapeutic or diagnostic potential is lacking. The increasing availability of whole-exome sequencing (WES) platforms has substantially improved the identification and investigation of the role of inherited and de novo variants in T1DM-associated genes. In this study, we implemented WES to identify variants in a cohort of 50 samples comprising 13 case–parent trios from singleton families with T1DM in the UAE. 

## 2. Materials and Methods

### 2.1. Study Participants and Recruitment

Thirteen families of UAE nationality were recruited, with at least one member clinically diagnosed with T1DM, as per the American Diabetic Association guidelines. There was a total of fifty individuals, in which 14 were T1DM patients. One family had two children with T1DM, while the rest only had one. Appendix A details the pedigree structure of the participants. All participants were provided with a questionnaire that included details on demographic information and medical history, such as age, gender, weight, height, waist circumference, systolic and diastolic blood pressure, family history of diseases, consanguineous marriages, and smoking status. For the 14 T1DM patients, biochemical tests were carried out, reporting glycosylated hemoglobin (HbA1c), hemoglobin, and white blood cell count (lymphocyte, monocyte). Saliva samples were collected from all subjects using the Oragene OGR-500 kit (DNA Genotek, Ottawa, ON, Canada). 

### 2.2. DNA Extraction and Library Preparation

Genomic DNA was extracted from the buccal cells in the saliva samples using the prepIT^®^L2P system (DNA Genotek, Ottawa, ON, Canada). The extracted DNA aliquots were quantified using the DS-11 FX Fluorometer (Denovix Inc. Wilmington, DE, USA), and the quality of each sample was assessed through agarose gel electrophoresis. Using the protocol recommended by the manufacturer of the Illumina TruSeq Exome Library Prep kit (Illumina Inc., San Diego, CA, USA), the libraries were prepared from the cleaned and sheared genomic DNA (gDNA). The indexed paired-end libraries were then quantified using the Denovix DS-11 FX Fluorometer to determine the optimal loading concentration of gDNA, providing the adequate clustering density on the flow cell during library sequencing. The fragment size was confirmed using the Advances Analytical Fragment Analyzer (Ankeny, IA, USA). Upon ensuring correct sizes and repeating incorrect runs, the libraries were loaded into NextSeq 500 (Illumina Inc., San Diego, CA, USA) separately.

### 2.3. Bioinformatics and Data Filtering Pipeline

The data analysis pipeline was designed based on the best practices recommended by the Broad Institute’s Genome Analysis Tool Kit (GATK) instructions, v4.0.6.0 [11]. After subjection to quality assurance, assuming default parameter, the raw reads of samples in the FastQ format files were checked using FastQC software, v0.11.5 [12]. The Trimmomatic tool version 0.33.0 was used to clip all the reads containing Illumina adapters with default parameters for paired-end sequencing. The reads were trimmed from the 3’ end [13]. Using the Burrows-Wheeler Aligner (BWA) v0.7.12 (BWA-MEM), alignment results were generated for each family member by mapping the raw reads to the human reference genome (GRCh37/hg19) [14,15] with reads of 76 base pairs (bps) in length. The sorted BAM files from all the sample lanes were merged, and the duplicate reads were removed using Picard version 2.9.4 tool’s commands SortSam, MergeSamFiles, and MarkDuplicates tools, respectively [16]. The samples’ mapping quality was checked using the Qualimap v2.2.1 tool and verified for an average coverage above 45X [17]. Using BaseRecaliberator from the GATK toolkit, the qualities of the mapped bases were improved, and the variant file from the dbSNP database was used to mark the sites of known variation [18]. 

The variant calling was performed using Haplotypecaller by GATK in GVCF mode. The results were jointly genotyped using the GenotypeGVCF tool, yielding the final raw variant file [19]. The GATK’s variant recalibration was performed to obtain both SNPs and INDELs before combining them into a single VCF file. Variant annotation was performed using snpEff v4.3p (i.e., frameshift, nonsynonymous, splice variant, etc.). SnpSift was used to annotate the variants using population allele frequencies from gnomAD v2.1.1, 1000 Genomes, and ExAC [20]. The resulting VCF file with the single-nucleotide variants/polymorphisms (SNVs/SNPs) and insertions or deletions (INDELs) that passed the GATK filtering were stored in a GEMINI (GEnome MINIng, v0.12.2) database after decomposing and normalization steps [21]. The variants were then loaded into the GEMINI database with the family’s pedigree file as the input file. The familial analysis was performed with the GEMINI de novo and autosomal recessive commands. Each variant was annotated by comparing it to several genome annotations from the GEMINI database, including the GnomAD, ENCODE tracks, UCSC tracks, OMIM, ClinVar, dbSNP, KEGG, and HPRD. The identified variants were then visually inspected using the IGV tool. The analysis was then focused on SNVs, SNPs, and INDELs that had minor allele frequency < 5% using gnomad_all, which we defined to be nonsynonymous, splice site, or missense SNPs. The impact information comes from snpEff through the GEMINI annotation that internally uses SIFT and PolyPhen to predict functional annotation. Only the variants that were repeated in two or more families were included in the next step of the analysis. Variants were classified as de novo if they were present in the affected child, but not in either of their parents.

### 2.4. Biological Functional Analysis

The clinical significance, disease associations, and linked phenotypes of the variants were determined using the ClinVar database, GeneCard, NCBI, and online databases [22]. Then, the search converged on genes with relations to type 1 diabetes and all its known pathways and related genes. The genome-wide association study (GWAS) of T1DM was cross-checked with all the variants before and after being filtered [23]. The Human Protein Atlas (http://www.proteinatlas.org (accessed on 1 December 2022)) and the Genotype-Tissue Expression v7 (GTEx) tool were utilized to understand the functional roles, regulatory landscape of gene expression, and splicing variation in a broad selection of primary human tissues. The NHGRI-EBI Catalog of human GWAS (www.ebi.ac.uk/gwas (accessed on 1 December 2022)) was used to identify if an SNP had been identified in a global population, or if they were distinctive to the Emirati population. 

## 3. Results

### 3.1. Demographic Factors

Of the 50 samples collected from the 13 families, there were 14 probands and 36 unaffected family members. All families had samples from an unaffected mother and unaffected father, and a subset (9 families) had samples of an unaffected sibling (Table 1). The mean age of the probands was 11.86 ± 5.08 years old, while the family members ranged from siblings to parents, and had a mean age of 31.46 ± 14.85 years old. There were nine males and five females in the proband group, whereas for the family members, 19 were male, and 17 were female. The average body mass index (BMI) of the probands was 16.94 ± 2.77, with a waist circumference of 64.92 ± 8.41, whereas the healthy family members had an average BMI of 27.46 ± 5.52, with a waist circumference of 87.82 ± 16.48. Nine out of thirteen families (69.2%) had consanguineous marriages.

Family history of the disease was also collected from the parents, and nine out of thirteen (69.23%) had a history of dyslipidemia, eight (61.54%) had a history of hypertension, five (38.46%) had a history of type 1 diabetes, and two (15.38%) had a history of hypothyroidism. The average HbA1c percentage was 9.12% ± 1.88%, which was higher than the normal range (<6%) for healthy individuals. The rest of the variables (hemoglobin: 13.13 ± 1.40; white cell count: 6.95 ± 4.00; lymphocytes: 5.87± 7.30; monocytes: 2.33 ± 3.58) were all within regular ranges and, thus, had no further significance. The lifestyle of the sample families was assessed for smoking habits, revealing 100% of the probands never smoked, and 25% of the family members never smoked. 

### 3.2. De Novo Variants

Only 1186 variants with high impact passed the filtering pipeline, as detailed in Appendix A. After eliminating the variants with less than 0.05 minor allele frequency (MAF) and variants that were not found in 2 or more families, we had 98 variants. Finally, we filtered out the genes with no biological relevance to T1DM or autoimmune pathways/diseases, and were left with 30 rare variants that belonged to 20 different genes (Table 2). The genes found to be related to T1DM were ten variants across seven genes (*MST1*, rs201139286; *TDG*, rs760400700, rs764159587; *TYRO3*, rs746533465, rs750893216, rs757748573; *IFIHI*, rs141469634; *GLIS3*, rs113076411; *VEGFA*, rs750060813; *TYK2*, 19-10475177-TA-T). There were twenty variants across thirteen genes that were linked to other forms of diabetes (type 2 diabetes (T2D), gestational diabetes, diabetes retinopathy, and nephrogenic diabetes insipidus) or other autoimmune diseases (*BCR*, rs372013175; *CACNA1B*, 9-140773611-G-GACGACACGGAGCCCTATTTCATCGGGATCTT; *CNN2*, 19-1036442-C-A; *COLGALT1*, 19-17666649-G-A; *LAMA3*, 18-21338476-T-G; *LGALS9C*, rs376412531; *MBD4*, 3-129155546-CT-C; *MST1L*, rs11260920; *MUC6*, rs368342230, rs376177791, rs754249101, rs761220536, rs766751467, rs766833662; *PABC1*, rs140822921; RNASEH2B, rs200320729; *ZNF596*, rs756701581).

The *MST1L* gene located in 1p36.13 is involved in the regulation of macrophage chemotaxis, and mainly expressed in the kidney, liver, and pancreas tissue. This will trigger an inflammatory response, which initiates insulitis and pancreatic cell death, leading to the production of IFNγ, TNFα, IL-1β, and the amplification of beta cell death cycle [24,25]. The *IFIH1* gene located in 2q24.2 is an innate immune receptor that plays a major role in sensing viral infection and in the activation of a cascade of antiviral responses, including the induction of type I interferons and proinflammatory cytokines. The *IFIH1* gene has been associated with the pathogenesis of diabetes (type 1 and type 2) and multiple autoimmune diseases [23,26,27,28,29,30,31,32,33,34,35,36]. *MBD4,* located in 3q21.3, is involved in DNA glycosylase and endonuclease activity, and is mainly expressed in lymphocytes. It is associated with obesity, BMI, sclerosis, cancer, and autoimmune disease [37,38,39,40]. 

The *MST1* gene located in 3p21.31 is involved in the regulation of T cell selection, and its deficiency restores normoglycemia, improves beta cell function and prevents the development of diabetes [24,27,28,41]. The *VEGFA* gene located in 6p21.1 is mainly expressed in thyroid tissue, and has been associated with severe retinopathy in type 1 diabetes, and glomerular microvasculature in diabetes, specifically due to islet vessel density, alteration in expression of genes regulating islet blood flow, insulin deficiency, and inflammation in intra-islet endothelial cells [42,43,44]. The *PAPBC1* gene located in 8q22.3 encodes a poly(A) binding protein and is associated with tumor progression [45,46]. The *ZNF596* gene located in 8p23.3 is highly expressed in the brain and cerebellum [47]. Characterized by lesions in the central nervous system disseminated in time and space [48], *ZNF596* has been reported to be involved in the pathogenesis of multiple sclerosis [49]. 

The *CACNA1B* gene located in 9q34.3 is involved in the N-type voltage-dependent calcium channel, which can cause beta cell dysfunction and death, and lead to both types 1 and 2 diabetes [50]. The *CACNA1B* gene has been associated with acute lymphoblastic leukemia and myeloid leukemia through the regulation of immune functions and leukocyte chemotaxis [51,52]. The *GLIS3* gene located in 9p24.2 encodes a nuclear protein that is involved in the expression and development of pancreatic beta cells, and has been associated with neonatal diabetes, fasting blood glucose, type 2 diabetes, and congenital hypothyroidism [53,54,55,56]. *GLIS3* plays a role in the generation of pancreatic beta cell viability and susceptibility to immune and metabolic-induced stress, such as proinflammatory cytokines and glucose oxidation [57]. The *MUC6* gene located in 11p15.5 is expressed in the stomach and pancreas tissue, is associated with hypertrophic cardiomyopathy, and is involved in enhancing innate immune reactivity [58]. Glycosylation of *MUC6* is found to upregulate the IL-17 response, which is found to be related to other immune-mediated diseases [59]. The *TYK2* gene located in 19p13.2 is associated with the cytoplasmic domain of type I and type II cytokine receptors, and is a component of both the type I and type III interferon signaling pathways [60,61]. The *TYK2* gene has a critical importance in the etiology of autoimmune and inflammatory diseases, specifically type 1 and type 2 diabetes, due to its association with the pancreatic β-cell-specific suppression of cytokine response including IFN [61,62,63,64,65]. 

The *TDG* gene located in 12q23.3 is induced by β cells and inflammatory mediators that play a key role in initiating the autoimmune response. Given that TDG enzyme control activates DNA demethylation, *TDG* gene expression was significantly upregulated in the IFN-α–treated islets and lymphocyte cells [66]. The *RNASEH2B* gene located in 13q14.3 is involved in the activation of the interferon pathway, leading to the infiltration of lymphocytes and mononuclear cells, and local chronic inflammation [67]. The *TYRO3* gene located in 15q15.1 regulates immunoregulation, plays an important role in the inhibition of the Toll-like-receptor-mediated innate immune response, and is an essential regulator of immune homeostasis [68,69]. *LGALS9C*, located in 17p11.2, is involved in cytoplasmic intracellular functions, and controls AMP-activated protein kinase in response to lysosomal damage, which is caused by diabetes, immune responses, and obesity [70]. This gene is also involved in attenuating T-cell expansion, tumor microenvironment, and chronic infections [71,72]. *LAMA3*, located in 18q11.2, is associated with the binding to cells via a high-affinity receptor through embryogenesis. While *LAMA3* is not associated with T1DM and is mainly expressed in lung tissue, it is associated with autoimmune diseases, immunologic isotypes, immune cytolytic activity, and ovarian cancer [73,74]. 

The *CNN2* gene located in 19p13.3 is involved in the structural organization of actin filaments, playing a role in smooth muscle contraction. While it is not associated with type 1 diabetes, it is expressed in fibroblast cells and linked to the innate immune system, myometrial relaxation, and contraction pathways [75,76]. *COLGALT1,* located in 19p13.11, encodes collagen β (1-O) galactosyltransferase 1 (ColGalT1), and is associated with musculoskeletal defects, cerebral small vessel disease, and congenital porencephaly [77,78,79]. While the *COLGALT1* gene is not associated with T1DM, it is correlated with autoimmune diseases as it could potentially antagonize the innate immune response [80]. The *BCR* gene in 22q11.23 acts as a GTPase-activating protein that encodes a novel serine/threonine kinase activity, and can be considered as a candidate tumor suppressor gene involved in meningioma pathogenesis and chronic myeloid leukemia [76,81]. The gene is expressed in the brain, endocrine, thyroid, and lymphoid tissue, which may trigger the intracellular signaling pathways leading to the expression of genes required for immune response [82,83]. 

## 4. Discussion

This study has identified genetic variants associated with the development of T1DM from 13 Emirati case–parent trios. Of the 13 families sampled for this project, 12 had reported de novo variants, with Family 7 having the highest number (nine) of variants linked to T1DM/autoimmune pathways, whilst Family 4 did not have any variants past the filtering steps. Two of the families (3 and 8) were consanguineously related by the third degree, and were the only families that were associated with the *CNN2* gene, linked to the innate immune system and autoimmune diseases. 

Gene *MUC6* had the highest number of variants (seven) associated with nine families, with two variants present in four families, two variants present in three families, and three variants present in two families. Gene *MUC6* has been identified as a loss of function (LoF) variant in the UAE population that is common in the local population (AF > 5%), and rare in global population (AF < 1%, as per the gnomAD exon catalog) [84]. *MUC6* has been associated with prostate carcinoma in the UAE population, hence promoting tumorigenesis [84,85]. The gene ontology annotations related to this gene include extracellular matrix structural organization, which may lead to glucose-induced endothelial damage and metabolic disturbances [86]. Sanger sequencing is recommended to identify the association of gene *MUC6* with T1DM in the Emirati population.

In comparison to global population, there were ten variants of seven genes reportedly associated with T1DM (*MST1*, rs201139286; *TDG*, rs760400700, rs764159587; *TYRO3*, rs746533465, rs750893216, rs757748573; *IFIHI*, rs141469634; *GLIS3*, rs113076411; *VEGFA*, rs750060813; *TYK2*, 19-10475177-TA-T). In addition to T1DM, the following genes were associated with other conditions from global GWAS analysis: the *MST1* gene was associated with multiple chronic inflammatory disease, including inflammatory bowel disease, Crohn’s disease, psoriasis, and ulcerative colitis [28,87]; the *TYRO3* gene was associated with the blood pressure, metabolic syndrome, and microalbuminuria [88,89,90]; the *IFIH1* gene was associated with psoriasis, systemic lupus erythematosus, chronic inflammatory diseases, autoimmune thyroid disease, and congenital hypothyroidism [23,26,27,28,29,30,31,32,33,34,35,36]; the *GLIS3* gene was associated with type 2 diabetes, chronic obstructive pulmonary disease, and asthma [56,91,92]; and the *TYK2* gene was associated with systemic lupus erythematosus, psoriasis, inflammatory bowel disease, rheumatoid arthritis, and type 2 diabetes [30,36,93,94]. 

While not associated with T1DM, there were twenty variants of thirteen genes that were linked to other forms of diabetes (type 2 diabetes (T2D), gestational diabetes, diabetes retinopathy, and nephrogenic diabetes insipidus) or other autoimmune diseases (*BCR*, rs372013175; *CACNA1B*, 9-140773611- G-GACGACACGGAGCCCTATTTCATCGGGATCTT; *CNN2*, 19-1036442-C-A; *COLGALT1*, 19-17666649-G-A; *LAMA3*, 18- 21338476-T-G; *LGALS9C*, rs376412531; *MBD4*, 3- 129155546-CT-C; *MST1L*, rs11260920; *MUC6*, rs368342230, rs376177791, rs754249101, rs761220536, rs766751467, rs766833662; *PABC1*, rs140822921; *RNASEH2B*, rs200320729; *ZNF596*, rs756701581), when compared to the global population. The following genes were associated with other conditions from GWAS analysis from global databases: the *CACNA1B* gene was associated with acute myeloid leukemia [51]; the *LAMA3* gene was associated with ovarian cancer [74]; the *MBD4* gene was associated with systemic sclerosis [38]; the *MUC6* gene was associated with hypertrophic cardiomyopathy and peptic ulcer disease [58,95]; and the *RNASEH2B* gene was associated with rheumatoid arthritis and prostatic hyperplasia [96,97]. Given that T1DM is a multifactorial autoimmune endocrine disease, several susceptible genes may be shared across different conditions [23,98,99,100]. These findings increase our understanding of the genetic contribution and biology underlying T1DM development, and suggest overlapping genetic origins with autoimmune disease and other forms of diabetes.

Several limitations affect the power of trio-based genetic analysis, including locus heterogeneity, sample size, pedigree structure, and genotype accuracy. Given the low sample size, widespread mutational recurrence and heterogeneity may be present, hence a larger sample size is required to confirm the de novo genetic findings. Family 5 had weak coverage and mapping quality in the father’s sample; therefore, variants identified in Family 5 should be considered with caution, which includes rs372013175 of *BCR*, 19-1036442-C-A of *COLGALT1*, rs200320729 of *RNASEH2B*, and rs746533465 of *TYRO3*. Additional functional studies in cell culture and animal models, as well as re-sequencing the identified genes in larger cohorts, are needed to assess the pathogenicity of the genes to T1DM. Since geographic isolation and consanguinity-driven genomic homozygosity may lead to the enrichment of founder mutations in specific ethnic groups, the presence of such rare mutations in our cohort from consanguineous families is not surprising. Therefore, further studies must be conducted in different ethnic groups to further understand the genetic landscape of T1DM.

## 5. Conclusions

We have identified de novo genes that may play a role in the pathogenesis of T1DM by conducting whole-exome sequencing on 13 trios from singleton Emirati families with T1DM. As per published GWAS, all the identified genes were either associated with T1DM or were associated with an autoimmune disease. The susceptibility loci for T1DM are heterogeneous, hence further work around identifying such genes may play a robust role in the pathogenesis of T1DM. Future studies can also assess the association of genes with different modes of inheritance, such as autosomal recessive, autosomal dominant, and X-linked. This can aid in understanding the linkage to disease pathogenesis on a wider spectrum, and may offer more relationships between non-HLA genes and T1DM among the citizens of the UAE. Finally, focused PCR-RFLP studies of particularly reported genes from GWAS studies on Emirati subjects would be beneficial to understand the impact of these genes in this region, and would allow the confirmation of variants reported in this study. With the findings presented in this paper, the next step ought to be broadening the spectrum of the search for more disease-causing variants that may be indicative of prediction of type 1 diabetes progression and etiology.

## Figures and Tables

**Table 1 biology-12-00413-t001:** Demographic and anthropometric data of probands and healthy family members.

Variable	T1DM (N = 14)	Healthy Family Members (N = 36)
Age (years)	11.86 ± 5.08	31.46 ± 14.85
Gender		
Male	9 (64.29%)	19 (52.78%)
Female	5 (35.71%)	17 (47.22)
BMI (kg/m^2^)	16.94 ± 2.77	27.46 ± 5.52
Waist circumference (cm)	64.92 ± 8.41	87.82 ± 16.48
Systolic blood pressure (mmHg)	108.43 ± 10.89	119.29 ± 14.15
Diastolic blood pressure (mmHg)	69.79 ± 7.85	74.14 ± 10.17

Continuous variables were recorded as mean ± standard deviation; categorical variables were recorded as number of events (percentages %); BMI: body mass index.

**Table 2 biology-12-00413-t002:** De novo genes with rare variants identified from 13 Emirati case–parent trios with T1DM.

Chr	Gene	Impact	rs-ID	Family ID	AF	Family Genotype	Link to T1DM	Link to Auto-Immune Diseases
1	*MST1L*	Stop Gained	rs11260920	Family 1 + 12	5.51 × 10^−2^	**G/A**,G/G,G/G,G/G,G/G	No	Yes
2	*IFIH1*	Intron Variant	rs141469634	Family 8 + 9	7.79 × 10^−3^	**CA/C**,CA/CA,CA/CA,CA/CA	Yes	Yes
3	*MBD4*	Frameshift Variant	rs747480541	Family 2 + 7 + 11	3.11 × 10^−3^	**CT/C**,CT/CT,CT/CT	No	Yes
3	*MST1*	Splice Acceptor Variant	rs201139286	Family 3 + 13	8.15 × 10^−3^	**T/G**,T/T,T/T,T/T	Yes	No
6	*VEGFA*	Intron Variant	rs750060813	Family 5 + 7	7.24 × 10^−4^	**T/C**,T/T,T/T	Yes	Yes
8	*PABPC1*	Frameshift Variant	rs140822921	Family 3 + 9	1.15 × 10^−3^	**C/CA**,C/C,C/C	No	Yes
8	*ZNF596*	Stop Gained	rs756701581	Family 2 + 7	3.98 × 10^−5^	**C/A**,C/C,C/C,C/C	No	Yes
9	*CACNA1B*	Splice Donor Variant	None	Family 1 + 9 + 10	9.04 × 10^−3^	**G/GACGACACGGAGCCCTATTTCATCGGGATCTT**,G/G,G/G,G/G	No	Yes
9	*GLIS3*	Synonymous Variant	rs113076411	Family 8 + 10	4.80 × 10^−3^	**CT/C**,CT/CT,CT/CT,CT/CT	Yes	Yes
11	*MUC6*	Frameshift Variant	rs368342230	Family 3 + 7 + 8	4.11 × 10^−6^	**TG/T**,TG/TG,TG/TG,TG/TG	No	Yes
			rs376177791	Family 3 + 7 + 8	1.01 × 10^−5^	**G/GT**,G/G,G/G,G/G		
			rs754249101	Family 6 + 9	1.87 × 10^−4^	**T/TGC**,T/T,T/T,T/T		
			rs761220536	Family 1 + 2 + 7 + 13	4.08 × 10^−6^	**G/GTGACGGT**,G/G,G/G,G/G		
			rs766751467	Family 6 + 9	7.30 × 10^−5^	**GCA/G**,GCA/GCA,GCA/.,GCA/GCA		
			rs766833662	Family 1 + 2 + 7 + 13	0.00	**CTGGTGCG/C**,CTGGTGCG/CTGGTGCG,CTGGTGCG/CTGGTGCG,CTGGTGCG/CTGGTGCG		
			rs776466780	Family 8 + 12	1.51 × 10^−4^	**G/GTA**,G/G,G/G,G/G		
11	*TYK2*	Protein Structural International Locus	None	Family 5 + 6	2.20 × 10^−4^	**TA/T**,TA/TA,TA/TA,TA/TA	Yes	Yes
12	*TDG*	Splice Donor Variant	rs760400700	Family 2 + 10	3.63 × 10^−2^	**T/G**,T/T,T/T,T/T	Yes	No
			rs764159587	Family 2 + 10	4.11 × 10^−2^	**G/GA**,G/G,G/.,G/G		
13	*RNASEH2B*	Frameshift Variant	rs200320729	Family 2 + 5	3.95 × 10^−2^	**T/TA**,T/T,T/T,T/T	No	Yes
15	*TYRO3*	Splice Donor Variant	rs746533465	Family 5 + 13	7.81 × 10^−5^	**G/GAGAGTTTGGTTCAGTGCGGGAGGCCCAGC**,G/G,G/G,G/G	Yes	Yes
			rs750893216	Family 1 + 6	1.37 × 10^−3^	**G/GGAGA**,G/G,G/G,G/G		
			rs757748573	Family 5 + 7 + 13	4.82 × 10^−4^	**G/C**,G/G,G/G,G/G		
17	*LGALS9C*	Frameshift Variant	rs376412531	Family 8 + 12	5.89 × 10^−2^	**GGA/G**,GGA/GGA,GGA/GGA,GGA/GGA	No	Yes
18	*LAMA3*	Splice Donor Variant	None	Family 1 + 7	NA	**T/G**,T/T,T/T	No	Yes
19	*CNN2*	Stop Gained	None	Family 3 + 8	NA	**C/A**,C/C,C/C,C/C	No	Yes
19	*COLGALT1*	Stop Gained	None	Family 5 + 11	NA	**G/A**,G/G,G/G,G/G	No	Yes
22	*BCR*	Frameshift Variant	rs372013175	Family 5 + 12	2.79 × 10^−3^	**T/TCCGG**,T/T,T/T,T/T,T/T	No	Yes

Chr: chromosome; AF: allele frequency; NA: not applicable; T1DM: type 1 diabetes mellitus. Allele frequency was obtained from the global population using gnomAD.broadinstitute.org. Family genotype is noted as follows, in chronological order: affected child, mother, father, sibling. Genotype of the affected child is in bold.

## Data Availability

All materials will be submitted in the Appendix A.

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
