# Peer review of "Whole-Exome Sequencing in Family Trios Reveals De Novo Mutations Associated with Type 1 Diabetes Mellitus"

_biology, 2023, doi:10.3390/biology12030413_

Round 1

Reviewer 1 Report

Mousa et al., identified de novo variants for Type 1 diabetes mellitus in 13 trio families. 

1. As the author mentioned 12 families had reported de novo variants. The detail of identified variants is mentioned in table 3. The references or sources of reported variants are not mentioned in the manuscript. It should be added in Table 3 or can be in the text as well. 

2. Did the author check the allele frequency of variants in other databases (bravo, all of us, if not it should be mentioned in the table as well.

3. As mentioned in the footnote of Table 3, NR = Not Reported but NR is not mentioned in table 3. 

4. On page 5 after Table 2, there is a detailed description of genes which can either be summarized in the discussion section rather than in the results section, to support the results. 

Author Response

Reviewer 1:

  1. As the author mentioned 12 families had reported de novo variants. The detail of identified variants is mentioned in table 3.The references or sources of reported variants are not mentioned in the manuscript. It should be added in Table 3 or can be in the text as well.

De novo in the context of family mainly refer to those mutations that are found in the proband but not the parents. The information in table 3 is from the annotation pipeline that is used by GEMINI tool, which incorporates the following tools: GnomAD, ENCODE tracks, UCSC tracks, OMIM, dbSNP, KEGG, and HPRD. The impact information comes from  snpEff through the GEMINI annotation that internally uses SIFT and PolyPhen to predict functional annotation. This has been added in the text. Thank you for bringing this to our attention.

  1. Did the author check the allele frequency of variants in other databases (bravo, all of us, if not it should be mentioned in the table as well.

Thank you for this comment. The allele frequency of variants in Table 3 is based on the global population using GnomAD. Given that there is no Emirati database that is publically available, we are unable to extract this information from the local population. This has been specified in the table.

  1. As mentioned in the footnote of Table 3, NR = Not Reported but NR is not mentioned in table 3.

Thank you very much for bringing this to our attention. “NR = Not Reported” has been removed from the footnote.

  1. On page 5 after Table 2, there is a detailed description of genes which can either be summarized in the discussion section rather than in the results section, to support the results.

Thank you for this suggestion. This comment has been taken into consideration and we have decided to expand on the description of genes, in comparison with other public datasets and populations. This will allow us to understand whether these variants are distinctive to the Emirati population. 

Reviewer 2 Report

Comments to Authors:

Mousa et al, explored the whole exome sequencing to determine the gene changes in parent-trio families to identify the T1D related genes that are passed on to the offspring resulting in their T1D Pathogenesis.

Overall article is well-written and self-explanatory. There are a couple of small clarifications that needs to be provided.

1.       It is mentioned in Results section, Page6, line 221 that TYK2 gene is responsible for autoimmune and inflammatory diseases and in Table-3; chr11, gene TYK2, it was mentioned that the gene is involved in T1DM. But in the discussion, Page 9- lines 276-278, it is said that TYK2 is associated with T2DM.  Can you clarify if that was a typing error, or does it have association with both T1 and T2DM?

2.       If the family 5 has weak coverage and mapping due to father’s sample, can you eliminate the results of that family? I know the n is low but removing family 5 doesn’t change the direction of your results.

Author Response

Reviewer 2:

  1. It is mentioned in Results section, Page 6, line 221 that TYK2gene is responsible for autoimmune and inflammatory diseases and in Table-3; chr11, gene TYK2, it was mentioned that the gene is involved in T1DM. But in the discussion, Page 9- lines276-278, it is said that TYK2 is associated with T2DM. Can you clarify if that was a typing error, or does it have association with both T1 and T2DM?

Thank you very much for bringing this to our attention. TYK2 gene is associated to both T1DM and T2DM – we have clarified this in the text.

  1. If the family 5 has weak coverage and mapping due to father’s sample, can you eliminate the results of that family? I know the n is low but removing family 5 doesn’t change the direction of your results.

While Family 5 had a weak coverage from the fathers sample, the trio-family did pass the quality control/assurance step, and have reported de novo variants. Therefore, given that we have included the family in the analysis, we did state this as a limitation.

Reviewer 3 Report

The authors collected 50 samples from 13 Emirati case-parent trios with respect to T1DM and found 12 novel genetic variants that might be associated to T1DM using in-silico approaches. Here some of my main comments:

1. The authors write about quality assurance of the data but do not mention the parameters used. A similar case rises where several programs have been used without indicating if we should assume default parameters or if some have some deviations. This is quite important if we want to maintain reproducibility.

2. Versions of databases are also important for both reproducibility and interpretation of results, there are several instances where the version would be informative.

3. I think that the text that reads “This study has uncovered novel genetic variants associated to the development of T1DM from 13 Emirati case-parent trios” is too strong given the lack of experimental validation. I would strongly suggest that at least some variants are validated with Sanger sequencing to give support to the in-silico list.

4. The raw sequencing data must be made available through one of the common portals.

5. Muc6 has been lifted as having the highest number of variants, however there is no further discussion. To my knowledge muc6 tends to be highly mutated in almost any kind of sample.

6. Paragraph 2 and 3 in the discussion reads as the results section and it doesn’t seem to add to the text. I think this is an excellent opportunity to make some comparisons with other public datasets and investigate if these de novo mutations are distinctive to the Emirati population.

Author Response

Reviewer 3:

  1. The authors write about quality assurance of the data but do not mention the parameters used. A similar case rises where several programs have been used without indicating if we should assume default parameters or if some have some deviations. This is quite important if we want to maintain reproducibility.

Thank you for this comment. We have elaborated on the parameters used in the text. In summary, default parameters were utilized if none are mentioned.

  1. Versions of databases are also important for both reproducibility and interpretation of results, there are several instances where the version would be informative.

Thank you. We have added the versions of the databases, respectively.

  1. I think that the text that reads “This study has uncovered novel genetic variants associated to the development of T1DM from 13Emirati case-parent trios” is too strong given the lack of experimental validation. I would strongly suggest that at least some variants are validated with Sanger sequencing to give support to the in-silico list.

Thank you for this comment. We completely agree that sanger sequencing remains the gold standard for the validation of NGS genetic variants. We have mentioned this as a limitation in our text.

  1. The raw sequencing data must be made available through one of the common portals.

Thank you for your comment. We have now uploaded our data to the European Genome-Phenome Archive (EGA). Once we receive the data accession code, we will submit it to the journal, prior to publication.

  1. Muc6 has been lifted as having the highest number of variants, however there is no further discussion. To my knowledge muc6 tends to be highly mutated in almost any kind of sample.

This is a great point. Interestingly, in our previous manuscript by Elbait et al. 2021, after conducting a Loss of Function analysis on the variants that are common in the UAE population (AF>5%) and rare elsewhere (i.e., less than 1% in all gnomAD population when using AFs from the gnomAD exon catalog), 15 variants were detected. Among those, MUC6 was identified to be associated to carcinoma of the prostate organ. This has been further elaborated in the discussion.   

  1. Paragraph 2 and 3 in the discussion reads as the results section and it doesn’t seem to add to the text. I think this is an excellent opportunity to make some comparisons with other public datasets and investigate if these de novo mutations are distinctive to the Emirati population.

Thank you for this comment. To identify if these mutations are identified in global populations, we have used multiple tools and search engines, including: NHGRI-EBI Catalog of human GWAS, Human Protein Atlas and the Genotype-Tissue Expression v7 (GTEx) tool. We have specified this in the methodology and discussion.

Round 2

Reviewer 3 Report

I appreciate the authors taking the comments into consideration